# A Deep Learning Network for Individual Tree Segmentation in UAV Images with a Coupled CSPNet and Attention Mechanism

Lujin Lv [1,2,3], Xuejian Li [1,2,3], Fangjie Mao [1,2,3], Lv Zhou [4], Jie Xuan [1,2,3], Yinyin Zhao [1,2,3], Jiacong Yu [1,2,3], Meixuan Song [1,2,3], Lei Huang [1,2,3] and Huaqiang Du [1,2,3,*]

[1]  State Key Laboratory of Subtropical Silviculture, Zhejiang A & F University, Hangzhou 311300, China; lvlujin@stu.zafu.edu.cn (L.L.); lixuejian@zafu.edu.cn (X.L.); maofj@zafu.edu.cn (F.M.); xuanjie@stu.zafu.edu.cn (J.X.); zhaoyinyin@stu.zafu.edu.cn (Y.Z.); yujiacong@stu.zafu.edu.cn (J.Y.); soongmayxuan@stu.zafu.edu.cn (M.S.); huangl@stu.zafu.edu.cn (L.H.)

[2]  Key Laboratory of Carbon Cycling in Forest Ecosystems and Carbon Sequestration of Zhejiang Province, Zhejiang A & F University, Hangzhou 311300, China

[3]  School of Environmental and Resources Science, Zhejiang A & F University, Hangzhou 311300, China

[4]  Research Center of Forest Management Engineering of State Forestry and Grassland Administration, Beijing Forestry University, Beijing 100083, China; zhou2200033@bjfu.edu.cn

*  Correspondence: duhuaqiang@zafu.edu.cn; Tel./Fax: +86-(571)-6374-6363

**Abstract:** Accurate individual tree detection by unmanned aerial vehicles (UAVs) is a critical technique for smart forest management and serves as the foundation for evaluating ecological functions. Existing object detection and segmentation methods, on the other hand, have reduced accuracy when detecting and segmenting individual trees in complicated urban forest landscapes, as well as poor mask segmentation quality. This study proposes a novel Mask-CSP-attention-coupled network (MCAN) based on the Mask R-CNN algorithm. MCAN uses the Cross Stage Partial Net (CSPNet) framework with the Sigmoid Linear Unit (SiLU) activation function in the backbone network to form a new Cross Stage Partial Residual Net (CSPResNet) and employs a convolutional block attention module (CBAM) mechanism to the feature pyramid network (FPN) for feature fusion and multiscale segmentation to further improve the feature extraction ability of the model, enhance its detail information detection ability, and improve its individual tree detection accuracy. In this study, aerial photography of the study area was conducted by UAVs, and the acquired images were used to produce a dataset for training and validation. The method was compared with the Mask Region-based Convolutional Neural Network (Mask R-CNN), Faster Region-based Convolutional Neural Network (Faster R-CNN), and You Only Look Once v5 (YOLOv5) on the test set. In addition, four scenes—namely, a dense forest distribution, building forest intersection, street trees, and active plaza vegetation—were set up, and the improved segmentation network was used to perform individual tree segmentation on these scenes to test the large-scale segmentation ability of the model. MCAN's average precision (AP) value for individual tree identification is 92.40%, which is 3.7%, 3.84%, and 12.53% better than that of Mask R-CNN, Faster R-CNN, and YOLOv5, respectively. In comparison to Mask R-CNN, the segmentation AP value is 97.70%, an increase of 8.9%. The segmentation network's precision for the four scenes in multi-scene segmentation ranges from 95.55% to 92.33%, showing that the proposed network performs high-precision segmentation in many contexts.

**Keywords:** individual tree detection; Mask R-CNN; urban forest; deep learning; UAV; attention mechanism

## 1. Introduction

Urban forests are composed of scattered trees, forest belts, and patches and are important components of urban ecosystems [1]. They play a crucial role in carbon sequestration, oxygen release, water cycling, soil conservation, and mitigating the urban heat island effect [2]. Individual tree crown detection is a fundamental technique for estimating parameters such as the tree crown width, diameter at breast height, canopy closure, height, and

biomass, which are important for characterizing the ecological functions of forests [3–6]. Determining the number of trees in urban forests is also important for government decision-making and administrative management [7]. This number can serve as a basis for forest inventory and carbon sequestration capacity assessment and can help promote sustainable urban development [8]. Satellite remote sensing has been instrumental in monitoring urban forest resources for a long time [2]. However, due to the fragmented distribution and heterogeneous underlying surface of urban forests, it is often difficult to accurately identify and detect individual tree crowns with the limited resolution of satellite remote sensing images [9].

Due to the consistent advancements observed in unmanned aerial vehicle (UAV) technology, using UAVs equipped with high spatial resolution sensors has become a flexible, fast, accurate, and cost-effective method to acquire images for individual tree detection [10,11]. High spatial resolution images obtained by UAVs contain rich and detailed information about ground objects, which can help us better understand the scale of individual tree crowns in urban forests [12].

Individual tree crown monitoring methods based on digital images obtained from UAVs include the local maximum value method, edge detection algorithms, watershed algorithms, and the region growing method [13]. Mohan et al. [14] utilized the local maximum value method to segment individual trees in private forests in Wyoming, USA with satisfactory results. Moreover, Jing et al. [15], Liu et al. [16], Bochkovskiy et al. [17], and Zhang et al. [18] applied an edge detection algorithm, watershed algorithm, U-Net network combined with a watershed algorithm, and region growing method, respectively, to achieve high-precision segmentation results. However, these methods have limitations in data utilization. First, these studies are usually based on small datasets and focused on small areas. Second, these algorithms mainly use individual bands as input images without fully tapping into all the available information in UAV images [19]. In addition, these methods are usually unsupervised methods, and different parameters need to be set according to different detection objects during processing, but the parameter settings depend on expert knowledge [20], a problem that can lead to low detection accuracy in complex urban forest areas. In summary, it is more difficult to apply digital image processing methods to complex urban forest areas [2], so more efficient and accurate individual tree segmentation algorithms need to be developed to meet the needs of the field.

In recent times, the domain of computer vision has witnessed the substantial utilization of deep learning and convolutional neural network (CNN) methodologies. These advanced techniques have found extensive applications, especially in intricate assignments like image classification, semantic segmentation, and object detection [19,21]. With the constant improvement of computing resources and algorithm networks, deep learning models have demonstrated outstanding performance and reliable capability. Deep learning is a novel approach to individual tree crown segmentation and detection that achieves end-to-end learning and prediction through the training of multilayer networks [22]. In traditional machine learning methods, manually designing features involves selecting and engineering specific attributes or properties from the input data that are believed to be relevant for the learning task [23]. However, manually designing features can introduce biases and errors for several reasons. Firstly, this process is often subjective and relies heavily on human expertise and domain knowledge. Secondly, creating and refining manual features can be a time-consuming process [24]. Lastly, manual feature design typically explores only a limited set of features, potentially missing crucial patterns or relationships that exist in the data [25]. However, a deep learning approach avoids the biases and errors that may occur when designing features manually, due to the fact that deep learning models learn to extract features and make predictions directly from the raw data without the need to predefine features that may be biased [26].

Currently, DL-based tree crown detection includes one-stage detection and two-stage detection [27]. One-stage networks typically output the position and category of each object in the image through a neural network [28], with the typical models including

You Only Look Once (YOLO) [29] and Single Shot MultiBox Detector (SSD) [21]. They have the advantages of a fast detection speed and requiring few computational resources, making them suitable for real-time applications [30]. For tree crown detection, Jintasuttisak et al. [31] used the YOLOv5 network to achieve the fast and accurate detection of palm trees, while Zheng et al. [21] used the SSD network for individual tree segmentation in Thai orchards, which greatly improved the accuracy compared to the traditional methods. Two-stage networks implement object detection in two stages. In the first stage, a region proposal network (RPN) is employed to produce regions that serve as potential candidates., and the second stage inputs these candidate regions into a classification network for further classification and localization operations [8]. Although the detection accuracy of two-stage networks is generally higher than that of one-stage networks, it comes at the cost of slower speeds due to the two network operations needed, and it introduces an additional time delay and overhead [32]. Each operation takes a certain amount of time to execute, and the cumulative time of all operations in both phases is added together [33]. This may result in a longer overall processing time than in a one-stage approach, where only one set of operations needs to be performed. The quintessential two-stage network model is the Faster Region-based Convolutional Neural Network (Faster R-CNN) [34]. For tree crown detection, Mubin et al. [35] used Faster R-CNN for oil palm tree detection and achieved high accuracy, while Xi et al. [36] used multispectral UAV images combined with an improved Faster R-CNN network for individual Ginkgo biloba tree detection on a campus, resulting in good accuracy.

The Mask Region-based Convolutional Neural Network (Mask R-CNN) is an instance segmentation network that extends Faster R-CNN, integrating object detection and semantic segmentation tasks [37]. It is capable of generating per-pixel binary masks for each instance of the detected objects within the bounding box, effectively performing detection and segmentation simultaneously [38]. Iqbal et al. [39] achieved 91% accuracy in detecting and segmenting coconut trees using the Mask R-CNN approach. Yu et al. [20] compared the Mask R-CNN with the watershed and local maximum methods for individual tree segmentation in young artificial forests, demonstrating a higher level of accuracy in Mask R-CNN. Zhang et al. [40] improved the Mask R-CNN network for the individual tree segmentation of broad-leaved and coniferous forests within an ecological public welfare forest and compared it with the U-Net and YOLOv3 networks, revealing the advantages of the improved Mask R-CNN network in individual tree segmentation.

Although the Mask R-CNN network has shown high accuracy in segmentation in some studies, it has primarily been applied to individual-type forests, street trees, or isolated trees with relatively uniform tree crown sizes, uniform tree species, and uniform distributions, which are easy to recognize [41,42]. However, the accuracy of the Mask R-CNN network is less satisfactory for urban forests with complex backgrounds, dense distributions of trees, diverse tree species, and varying tree crown sizes and shapes [43,44]. In this paper, we propose an improved Mask R-CNN network named the Mask-CSP-attention-coupled network (MCAN). We introduce Cross Stage Partial Net (CSPNet) [45] in the backbone feature extraction network and add a channel–spatial attention mechanism in the feature pyramid network and use a new learning rate decay method and activation function to address the issue of low accuracy in individual tree segmentation in urban forests. The proposed method effectively reduces the missed detections of small targets and the false detections of objects in complex backgrounds, enabling the intelligent extraction of urban forest features from UAV imagery.

## 2. Materials and Methods

### 2.1. Study Area

The study area is situated in Zhejiang A & F University's East Lake campus, which is in Lin'an District, Hangzhou City, Zhejiang Province, as depicted in Figure 1. Over 3000 different species of subtropical evergreen vegetation, including ginkgo, willow, osmanthus, camphor, magnolia, and metasequoia, can be found in the area, which has a

subtropical monsoon climate. These plants are distributed in the study area in various forms, such as isolated trees, roadside trees, stands, and plantations. We conducted thorough tests for four different scenarios, including a dense forest distribution, building forest intersection, street trees, and active plaza vegetation, as shown in Figure 1, to gauge MCAN's ability to recognize tree crowns.

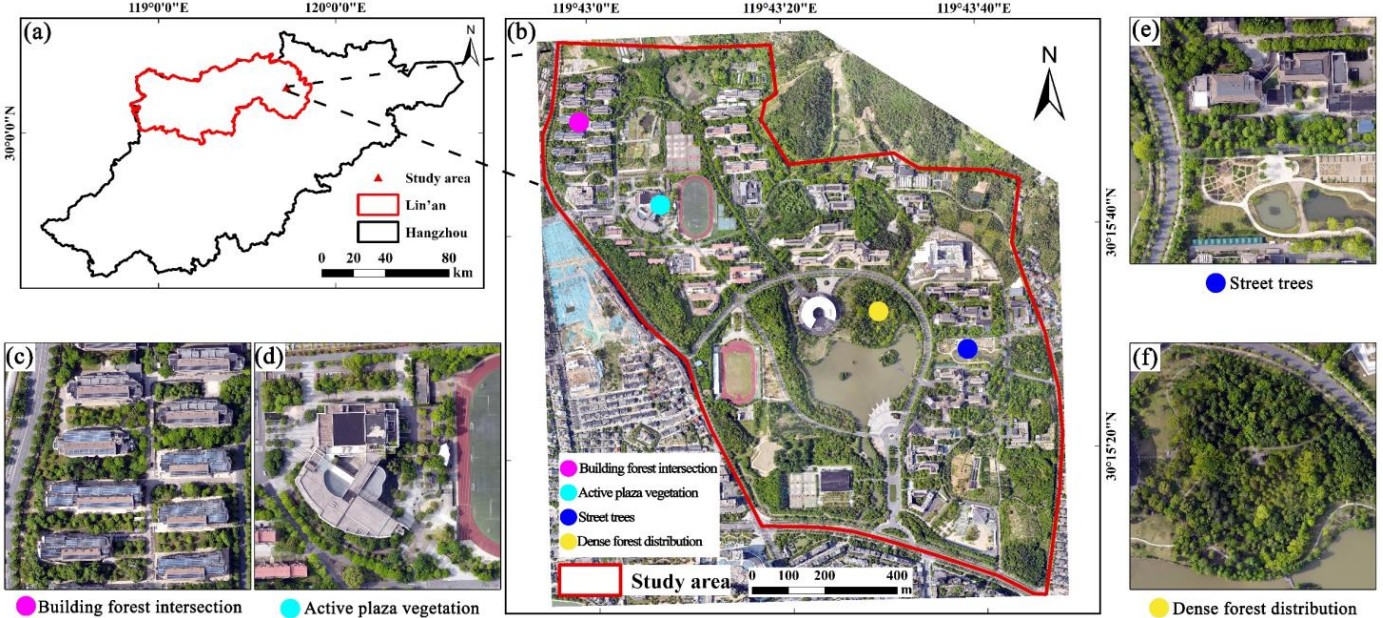

**Figure 1.** Overview of the study area. (**a**) Location of the study area, (**b**) UAV images of the study area, and (**c–f**) images of the four different scenes.

### 2.2. Datasets and Processing

The study used photos taken on 2 May 2022 using a DJI Mavic Air 2 drone with a 1/2-inch CMOS camera. The day of shooting was midday; the weather was clear, cloudless, and windless; and the sunlight intensity was stable. With a pitch angle of $-90°$, viewing angle of $84°$, an equivalent focal length of 24 mm, and an aperture of f/2.8, the drone was flown at a height of 300 m. The maximum ascent speed of the UAV was 4 m/s, the maximum descent speed was 3 m/s, the maximum flight time was 34 min, and the maximum range was 18.5 km. The images had a spatial resolution of approximately 0.1 m and consisted of three widely used spectral bands or RGB. The flight was programmed with an 80% overlap rate in the heading direction and a 70% overlap rate on the sides.

We used PIX4Dreact 1.4 software to process the images acquired by the UAV. Firstly, due to the high overlap rate of the images and the pitch angle of $-90°$ during the flight, we could obtain the corresponding points between the photos and then align the photos through the corresponding points to merge them to obtain the orthophotos. The size of the final orthophoto was 16,051 × 14,685, with a spatial resolution of 0.1 m, and the coordinate system was CGCS2000.

### 2.3. Introduction to Object Detection Algorithms

#### 2.3.1. YOLOv5

YOLOv5 is a fast and accurate one-stage target detection algorithm, which is the fifth-generation version of the YOLO series [46]. YOLOv5 has improved upon YOLOv4 [17], resulting in an elevated detection accuracy and accelerated processing speed. The structure of YOLOv5 includes three parts: backbone, feature pyramid network (FPN), and head [27]. Among them, the backbone utilizes Cross Stage Partial Dark Net (CSPDarkNet) [27], while the mosaic data enhancement method [47] and focus structure are added for image enhancement and channel expansion. In addition, in the FPN, YOLOv5 performs multiscale feature

fusion by up-sampling and down-sampling the feature maps output from the backbone [48]. The head module of YOLOv5 contains three branches, each of which is responsible for predicting target frames at different scales. In the head, the input features are reshaped into a 3D tensor that contains the position coordinates and category probabilities of the predicted target frames. These tensors are processed with Non-Maximum Suppression (NMS) [49] to output the final detection results.

### 2.3.2. Faster R-CNN

Faster R-CNN, derived from the frameworks of R-CNN and Fast R-CNN, is a two-stage object detection network [50]. Faster R-CNN utilizes VGG16 [51] for feature extraction; after which, the RPN is utilized to generate anchors and filter the suggestion frames [52]. The selected candidate regions are then fed into a target classification network for target classification and location regression. In this process, Faster R-CNN utilizes the region of interest pooling (ROI pooling) [53] operation to fix the different sizes of proposals to the same size, which facilitates the convolutional network operation.

### 2.3.3. Mask R-CNN

Mask R-CNN includes the backbone network, RPN, ROI Align, and classification section [37]. There are several steps in the Mask R-CNN network detection procedure. First, in the backbone, Resnet and FPN are used to extract multiscale feature data. Then, the ROI is generated using the shared convolutional layer of the RPN and fed into the later target classification and mask prediction network for further processing. Then, the ROI feature maps are extracted using the ROI Align part, and the proposals generated by the RPN network are pooled to fix the feature maps of different scales into a uniform scale for the convolutional network to predict the categories. Finally, fully connected networks (FCNs) are used to obtain the detection categories and bounding boxes while achieving semantic level segmentation.

### *2.4. MCAN*

### 2.4.1. The Overall Framework of MCAN

Figure 2 depicts the MCAN urban forest individual tree identification and segmentation model. The network replaces the backbone of the Mask R-CNN with a Cross Stage Partial Residual Net (CSPResNet) and also utilizes Sigmoid Linear Unit (SiLU) as an activation function and incorporates a channel–spatial attention module in the FPN.

### 2.4.2. CSPResNet

MCAN's backbone feature extraction network, CSPResNet, incorporates CSPNet. CSPNet consists of the convolutional framework module Conv Batch Normalization SiLU (CBS) and Bottleneck module, which divides the input features into the same two parts and improves the model accuracy by adding skip connections of the residual blocks, which alleviates the gradient disappearance problem and overfitting associated with increasing the depth of deep neural networks (see the CSPLayer (Cross Stage Partial Layer) module in the figure). CSPResNet incorporates the focus layer and spatial pyramid pooling (SPP) module. The focus layer is a special convolutional layer for the target detection task, mainly used for small target detection. The framework first splits a large input image into 4 copies and then performs convolutional operations on these 4 copies to extract the features of interest. By pooling the input features three times with different size pooling kernels and stacking them with the original features, the SPP module is able to increase the perceptual field of the feature network and effectively separate the most significant contextual features with little or no slowdown in the network operations [17]. The SiLU activation function [54], which is an improved version of the sigmoid activation function and the ReLU activation function and is now widely used in deep learning, is shown in Equation (1).

$$f(x) = x * \sigma(x) = x * \frac{1}{1 + e^{-x}} \tag{1}$$

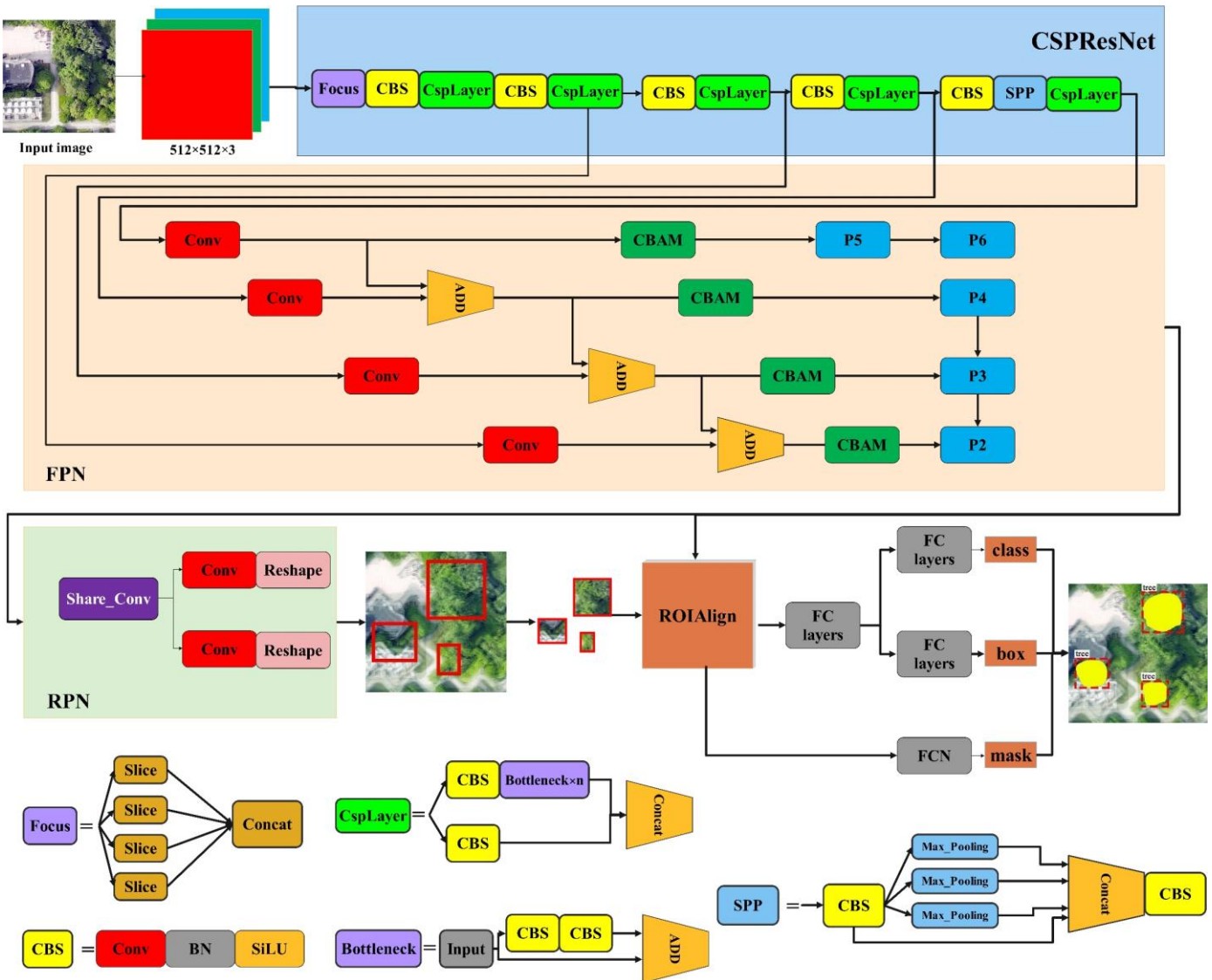

**Figure 2.** Overall framework of the MCAN model.

### 2.4.3. Channel–Spatial Attention Module

This study integrated an attention mechanism module into FPN to increase the precision of individual tree detection in complex environments. The attention mechanism enhances the model's performance and interpretability by helping it better focus on the key elements. The two basic types of attention mechanisms are channel attention (CA) and spatial attention (SA) [55]. CA refers to the channel dimensions of the input features and assigns different weights according to the importance of each channel [27]. It is generally used with operations such as pooling to generate the importance ranking of channels and then performs the convolution operation with the input features. SA refers to giving different weights to different spatial locations in a feature map, which is generally generated using a CNN, and then convolving with the input features.

As indicated in Figure 3, the convolutional block attention module (CBAM) [56] was employed in this investigation. This module consists of a channel attention module (CAM) and a spatial attention module (SAM). The channel attention weight is denoted by $M_c$, and the spatial attention weight is denoted by $M_s$.

**Convolutional Block Attention Module**

Figure 3. Overall framework of the CBAM.

The CBAM first uses the CAM to compute channel attention weights on input feature F to obtain the weight vector $M_c$ and performs the convolution operation of $M_c$ with input feature F to obtain F1; after which, F1 is used as the input of the SAM to compute spatial attention weights $M_s$ and performs the convolution operation of $M_s$ with F1 to obtain the output feature F.

The CAM compresses the spatial dimensions of the input features by MaxPool and AvgPool, sends them to the shared multilayer perceptron (MLP) for two full-connection operations, and obtains the channel weights $M_c$ using ADD and activation function operations. The CAM framework is shown in Figure 4 and Equation (2).

**Channel Attention Module**

Figure 4. CAM framework.

$$M_c(F) = \sigma(MLP(AvgPool(F)) + MLP(MaxPool(F))) \tag{2}$$

where $\sigma$ is the Softmax activation function, AvgPool and MaxPool are the average pooling operation and the maximum pooling operation, respectively, and MLP is the shared fully connected operation.

Additionally, SAM first pools the input features using MaxPool and AvgPool to compress the channel dimensions and stack the results, respectively, before performing the convolution operation with the calculation of the activation function to produce the spatial attention weights $M_s$. The SAM framework is shown in Figure 5 and Equation (3).

**Spatial Attention Module**

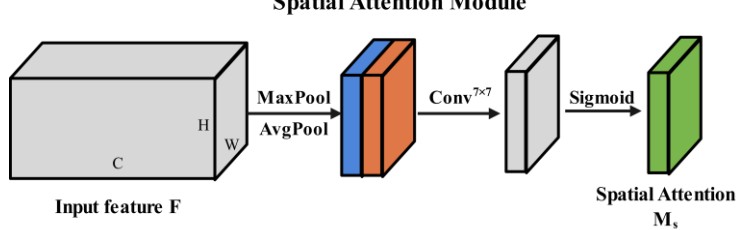

Figure 5. SAM framework.

$$M_s(F) = \sigma\Big(Conv^{7\times7}(Concat[AvgPool(F), MaxPool(F)])\Big) \tag{3}$$

where $Conv^{7\times7}$ represents a convolution operation employing a kernel size of $7 \times 7$.

### 2.5. Dataset Production

To facilitate the training of the deep learning model, we cropped the preprocessed images and excluded multiple scene prediction region ranges. In the end, we obtained 967 images with a resolution of $512 \times 512$ pixels. The 967 cropped photos were supplemented to create a dataset size of 3868 to increase the diversity of the sample data and lower the possibility of overfitting of the network model [57]. The enhancement methods included changing the brightness, rotation, and contrast of the image, as well as adding noise. To circle the canopy area and create example segmentation labels, we performed canopy contouring and labeling of the images based on visual interpretation and the labelme tool. We separated the dataset into a training set, a validation set, and a test set according to a ratio of 6:2:2. The tree crowns of some trees are totally hidden by the upper branches and foliage, rendering them unrecognizable; hence, they are not included in the label manufacturing. The total number of labeled individual woods in the expanded dataset is 56,020. The statistics of the number of individual trees for these data are shown in Table 1, while some of the sample images can also be seen in Figure 6.

**Table 1.** Statistics on the number of individual trees in the dataset.

| Dataset | Number of Individual Trees |
|---------|---------------------------|
| Training | 32,840 |
| validation | 12,110 |
| Test | 11,070 |
| Total | 56,020 |

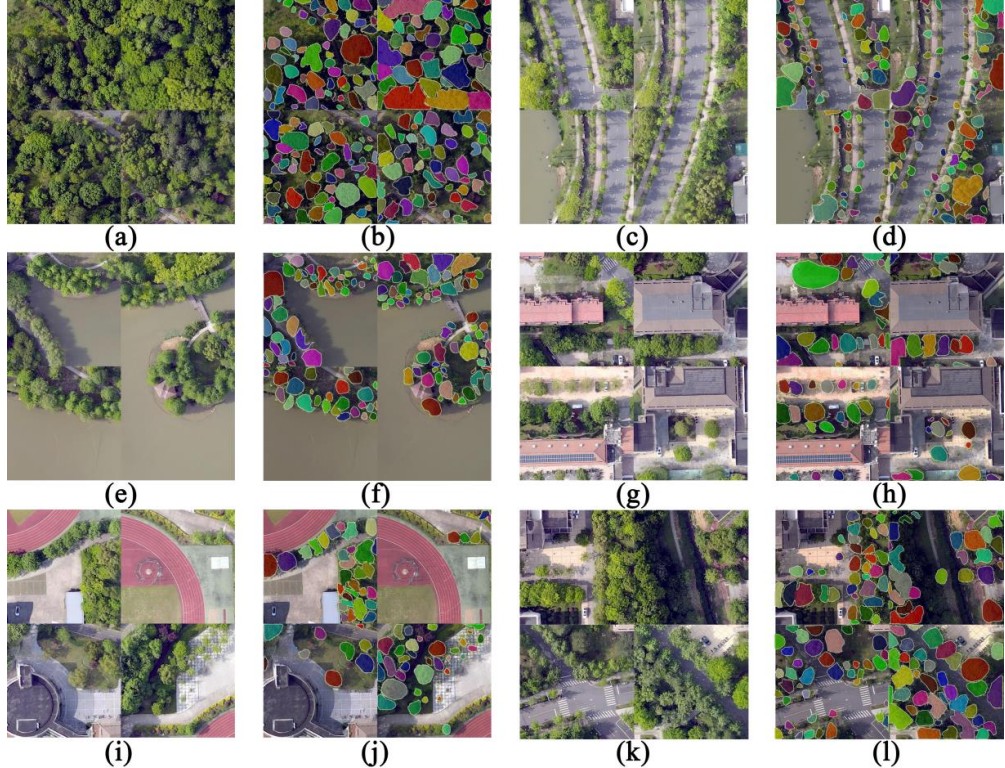

(a)   (b)   (c)   (d)

(e)   (f)   (g)   (h)

(i)   (j)   (k)   (l)

**Figure 6.** Schematic diagram of the sample with (**a**) dense forest, (**c**) street trees, (**e**) a watershed, (**g**) buildings, (**i**) activity sites, and (**k**) street trees. (**b,d,f,h,j,l**) The corresponding labels.

*2.6. Loss Function and Training Experiment*

The loss function of the MCAN network consists of two parts, $L_{RPN}$ and $L_{FPN\ head}$, where $L_{RPN}$ includes the classification loss $L_{rpn_{cls}}$ and target box offset loss $L_{rpn_{bbox}}$, and $L_{FPN\ head}$ includes the classification loss $L_{head_{cls}}$, target box offset loss $L_{head_{bbox}}$, and mask loss $L_{mask}$ [58]. The overall formula and each loss formula are shown in Equations (4)–(15).

$$L = L_{RPN} + L_{FPN\ head} \tag{4}$$

$$L_{RPN} = L_{rpn_{cls}} + L_{rpn_{bbox}} \tag{5}$$

$$L_{FPN\ head} = L_{head_{cls}} + L_{head_{bbox}} + L_{mask} \tag{6}$$

$$L_{rpn_{cls}} = \frac{1}{N_{rpn_{cls}}}\sum_i L_{cls}(p_i, p_i^*) \tag{7}$$

$$L_{rpn_{bbox}} = +\lambda_1 \frac{1}{N_{rpn_{bbox}}}\sum_i p_i^* L_{reg}(t_i, t_i^*) \tag{8}$$

$$L_{head_{cls}} = \frac{1}{N_{head_{cls}}}\sum_i L_{cls}(p_i, p_i^*) \tag{9}$$

$$L_{head_{bbox}} = \lambda_2 \frac{1}{N_{head_{bbox}}}\sum_i p_i^* L_{reg}(t_i, t_i^*) \tag{10}$$

$$L_{mask} = \gamma \frac{1}{N_{mask}}\sum_i L_{mask}(s_i, s_i^*) \tag{11}$$

Among them,

$$L_{cls}(p_i^*, p_i) = -\log[p_i^* p_i + (1 - p_i^*)(1 - p_i)] \tag{12}$$

$$L_{reg}(t_i, t_i^*) = \mathrm{smooth}_{L1}(t_i - t_i^*) \tag{13}$$

$$L_{reg}(t_i, t_i^*) = \begin{cases} 0.5(t_i - t_i^*)^2 & \mathrm{if}|t_i - t_i^*| < 1 \\ |t_i - t_i^*| - 0.5 & \mathrm{otherwise} \end{cases} \tag{14}$$

$$L_{mask}(s_i^*, s_i) = -(s_i^* \log s_i + (1 - s_i^*)\log(1 - s_i)) \tag{15}$$

where i represents the anchor frame, $p_i$ represents the probability that the ith anchor frame is predicted to be the target, and $p_i^*$ represents whether the ith anchor frame is the target ($p_i^* = 1$ when the anchor is the target, and $p_i^* = 0$ when the anchor is the background); $t_i$ represents the four parameterized coordinates obtained from the prediction of the ith anchor frame; $t_i^*$ represents the four parameterized coordinates corresponding to the real frame; and $\lambda_1$, $\lambda_2$, and $\gamma$ are parameters for balancing the number of anchor boxes $N_{cls}$ and the number of bounding boxes $N_{reg}$.

The operating system of this network training environment is Windows 10 Professional, the processor is Intel core i5-12400, the RAM capacity is 16 GB, and the NVIDIA GeForce GTX 3060 (12 GB) graphics card is used. The Python version is 3.7, and the deep learning framework is TensorFlow2.4.0-gpu.

To compare the performance difference after the model improvement, a comparison with other classical models is necessary. YOLOv5 and Faster R-CNN are both classical models for target detection and have also achieved good accuracy in individual tree detection [31,59,60], so this study compares MCAN with Mask R-CNN, YOLOv5, and Faster R-CNN. The training dataset, training parameters, and hardware configurations

were set consistently while keeping the same number of training rounds for all models to evaluate the differences for each metric. To save model training time, avoid the time required to train the network from scratch, and improve the training accuracy, this study conducted model training based on a migration learning approach. Transfer learning is a method that uses information obtained from a previous task, such as data features and model parameters, to help learn for a new task. This approach reduces the cost of training data and increases the efficiency of model application [61]. All four network models were trained using pretrained weights on the COCO dataset [62] as the initialization weights, while cosine annealing [63] was used to optimize the learning rate. Specifically, the updated learning rate $new_{lr}$ was calculated as shown in (16).

$$new_{lr} = eta_{min} + (initial\_lr - eta\_min) * \left( \frac{1 + \cos\left( \frac{cur_{epoch}}{T\_max} * \pi \right)}{2} \right) \quad (16)$$

The predetermined parameters include the initial learning rate initial_lr, the minimum learning rate eta_min, and the maximum number of epochs T_max. The value that represents the quantity of training rounds completed to date is the current training epoch number $cur_{epoch}$, the evaluation of each model's performance in terms of metrics pertaining to the detection of individual trees. The detailed parameters of the network are shown in Table 2.

**Table 2.** Network detailed parameters table.

| Parameters | Network | | | |
|---|---|---|---|---|
| | **MCAN** | **Mask R-CNN** | **Faster R-CNN** | **YOLOv5** |
| Optimizer | Adam | Adam | Adam | Adam |
| Initial learning rate | 0.001 | 0.001 | 0.001 | 0.001 |
| Momentum | 0.937 | 0.937 | 0.937 | 0.937 |
| Learning rate decay mode | Cosine annealing | Cosine annealing | Cosine annealing | Cosine annealing |
| Weight decay | 0.0005 | 0.0005 | 0.0005 | 0.0005 |
| epoch | 100 | 100 | 100 | 100 |
| batch size | 16 | 16 | 16 | 16 |
| Backbone network | CSPResNet | Resnet101 | Resnet50 | CSPDarkNet |
| Activation function | SiLU | SiLU | SiLU | SiLU |

*2.7. Accuracy Assessment*

The accuracy of individual tree detection and segmentation was assessed individually in this study to compare the quantitative performance of various algorithms. By comparing them to the expected outcomes of the models in the test set using the accurate annotation of the UAV pictures, the positions and contour ranges of individual trees were assessed. Recall, precision, F1 score, and average precision (AP) were the four measures that were employed in this study to assess each model's strengths and flaws. The most fundamental accuracy evaluation metrics for classification problems are precision and recall. Precision measures the proportion of detected objects that are actually objects, and recall measures the proportion of all objects correctly identified. To evaluate the precision of object detection in a particular category, the AP is one of the assessment metrics most frequently used in target detection. The precision–recall curve is used as the foundation for calculating the AP. The area under the precision–recall curve for each category is determined as the AP. This calculation's formula is displayed below, where TP stands for the number of correctly recognized individual trees, FN for the number of undetected individual trees, and FP for the number of mistakenly detected individual trees. The formulas are defined as shown

in (17)–(20) These measurements enable a more scientific and unbiased assessment of the performances of various models.

$$\text{Precision} = \frac{\text{TP}}{\text{TP} + \text{FP}} * 100\% \tag{17}$$

$$\text{Recall} = \frac{\text{TP}}{\text{TP} + \text{FN}} * 100 \tag{18}$$

$$\text{F1 Score} = 2 * \frac{\text{precision} * \text{recall}}{\text{precision} + \text{recall}} * 100\% \tag{19}$$

$$\text{AP} = \int_0^1 \text{p(r)dr} \tag{20}$$

## 3. Results

### 3.1. Model Training Results and Accuracy Evaluation

After 100 iterations, Figure 7 displays the loss curves of the training set for the four models MCAN, Faster R-CNN, YOLOv5, and Mask R-CNN, as well as the comparison of the PR curves on the test set following network convergence. The loss curves show that the loss values of the four models start off high during training and subsequently oscillate downward as the number of training epochs rises. Faster R-CNN has the slowest convergence pace among them, while the Mask R-CNN network and MCAN decrease faster during the initial training phase and reach convergence after 20 and 60 rounds, respectively. Faster convergence rates may result in fast learning and prediction capabilities but may also lead to overfitting [64]. Slower convergence rates may improve model stability and reduce the risk of overfitting [43]. Analysis of the PR curves shows that, when the recall is higher than 0.35, the precision rates of MCAN are all higher than the remaining networks, and the curves of MCAN are fuller and the best.

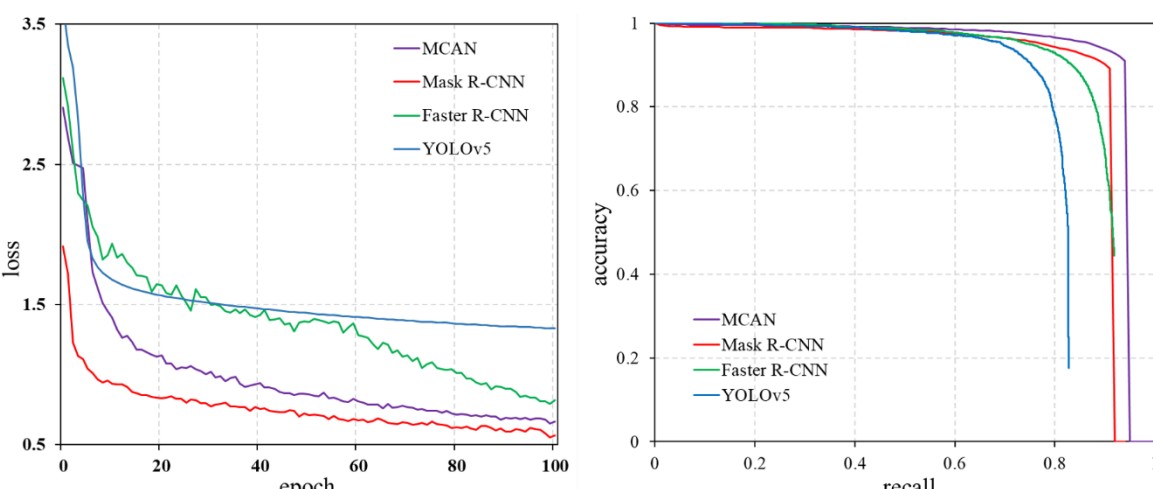

**Figure 7.** Plot of the model loss curve vs. PR curve.

Each trained model is used for individual tree detection in the test dataset when the training procedure is complete. The precision rate, recall rate, F1 value, and AP metrics of an individual tree in the test set under each model were computed based on the label production outcomes and field survey data. The confidence level used for this accuracy assessment metric was 0.5, and the IoU threshold range for calculating the AP was 0.5–0.95 with a step size of 0.05. Table 3 displays the accuracy comparison results of the four networks on the test set.

**Table 3.** Comparisons of model detection accuracy.

| Model | Recall | Accuracy | F1 Score | AP (@.5:.95) |
|---|---|---|---|---|
| YOLOv5 | 69.26% | 95.22% | 80.19% | 79.87% |
| Faster RCNN | 71.80% | 89.01% | 79.48% | 88.56% |
| Mask R-CNN | 72.40% | 96.14% | 82.60% | 88.70% |
| MCAN | 75.74% | 97.84% | 87.48% | 92.40% |

The information in Table 3 indicates that MCAN outperforms the other three networks in terms of detection outcomes. The recall rate is above 75%, the accuracy rate is above 95%, the F1 value is above 85%, and the AP is above 90%. The recall, accuracy, F1 and AP values of the other three networks, which were all below those of MCAN, ranged from 69% to 53%, 89% to 97%, 79% to 89%, and 79% to 89%, respectively.

### 3.2. Comparison of Individual Tree Segmentation Results

The segmentation accuracy metrics on the test sets of both networks were calculated to compare the benefits and drawbacks of MCAN and Mask R-CNN for segmenting individual trees, and the results are displayed in Table 4. With an 8.9% improvement in the AP, a 10.9% improvement in recall, a 0.82% improvement in accuracy, and a 7.23% improvement in the F1 score, MCAN is superior in all the accuracy evaluation criteria in Table 4.

**Table 4.** Comparison of model segmentation accuracy.

| Model | Recall | Accuracy | F1 Score | AP |
|---|---|---|---|---|
| Mask R-CNN | 62.60% | 87.22% | 72.89% | 88.80% |
| MCAN | 73.50% | 88.04% | 80.12% | 97.70% |

As shown in Figure 8, MCAN has higher feature detection and differentiation abilities for trees and other locations compared to Mask R-CNN. It also minimizes the amount of non-forest or shrub objects that are mistakenly classified as individual trees. By using an attention mechanism, MCAN enables the reduction of both the number of instances in which multiple individual trees are mistakenly identified as individuals and the number of instances in which individual trees are mistakenly identified as multiple trees.

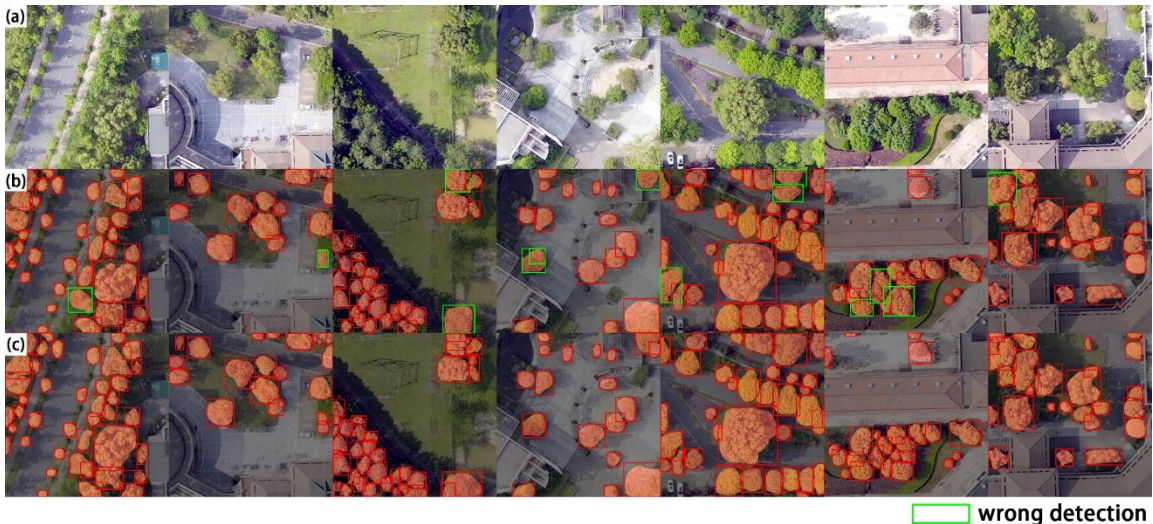

☐ **wrong detection**

**Figure 8.** Comparison of wrong detections. (**a**) The study images, (**b**) Mask R-CNN, and (**c**) MCAN.

By including CSPNet in the backbone feature extraction network, MCAN also increases the accuracy of small item recognition. This decreases the number of missing individual

trees and raises the recall rate. As seen in Figure 9, Mask R-CNN misses the solitary trees in the image with small crown sizes, whereas MCAN is able to identify them. Due to their similar colors and few pixels in comparison to the background, these missing individual trees are readily disregarded, which lowers the recall rate of the network.

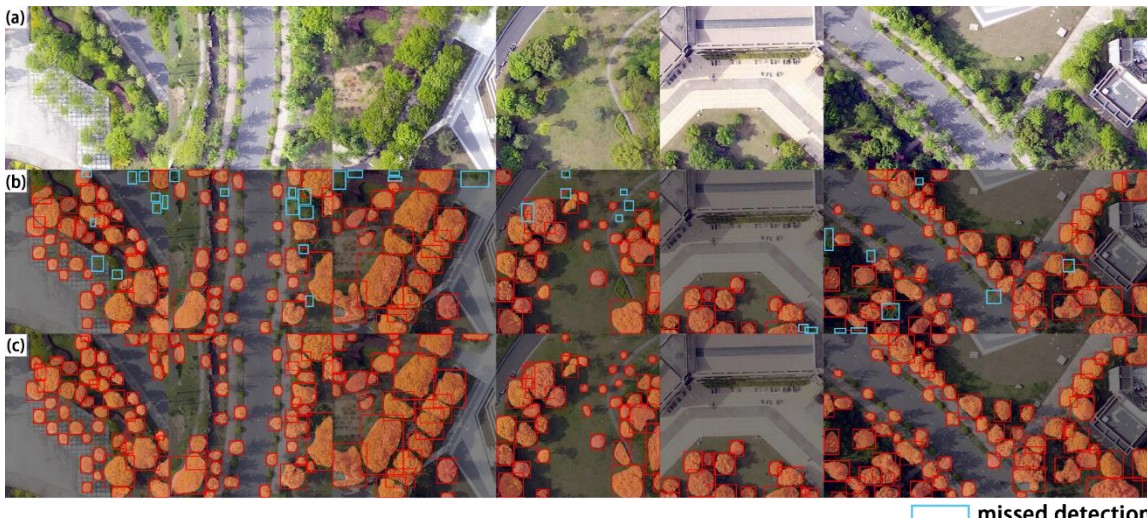

**Figure 9.** Comparison of missed detections. (**a**) The study images, (**b**) Mask R-CNN, and (**c**) MCAN.

### 3.3. MCAN in Multi-Scene Urban Forest Canopy Segmentation Application

We use MCAN to detect canopies in four different scenes on campus—namely, a dense forest distribution, building forest intersection, street trees, and active plaza vegetation—to fully verify the effectiveness and multi-scene performance of MCAN in large-scale canopy segmentation detection in urban forests. First, the scene area images are cropped to images $512 \times 512$ in size while maintaining a 10% overlap rate and sent to the model for prediction; after which, the prediction results are stitched together, and the duplicate detection areas are removed according to NMS. The prediction results are shown in Figure 10 and Table 5. The total pixel size of the four regions is 4096 pixels $\times$ 4096 pixels, and the spatial resolution is approximately 0.1 m. The prediction results can allow us to understand the segmentation statuses of different scales, different backgrounds, and different shapes of individual trees.

The analysis in Figure 10 and Table 5 shows that the MCAN model achieved good results in canopy detection in campus urban forests. Out of the total 1863 individual woods detected, the model detected 1850 individual woods, of which 119 individual woods were mistakenly identified as targets. In addition, the number of correctly detected individual woods was 1731, and the number of missed individual woods was 132. In addition, 1731 individual trees were accurately detected, whereas 132 individual trees were overlooked. MCAN's recall and precision rates in the four campus settings that were chosen were both above 90%. Scenario a has the highest precision rate among the four scenarios at 95.55%, while scenario b has the highest recall rate among the four scenarios at 94.62%. The other two scenarios also both achieved an accuracy rate of 92% and a recall rate of 90%. This shows that the MCAN has a good generalization ability and prediction capability to adapt to different scenarios and can be effectively used for canopy detection tasks over large areas.

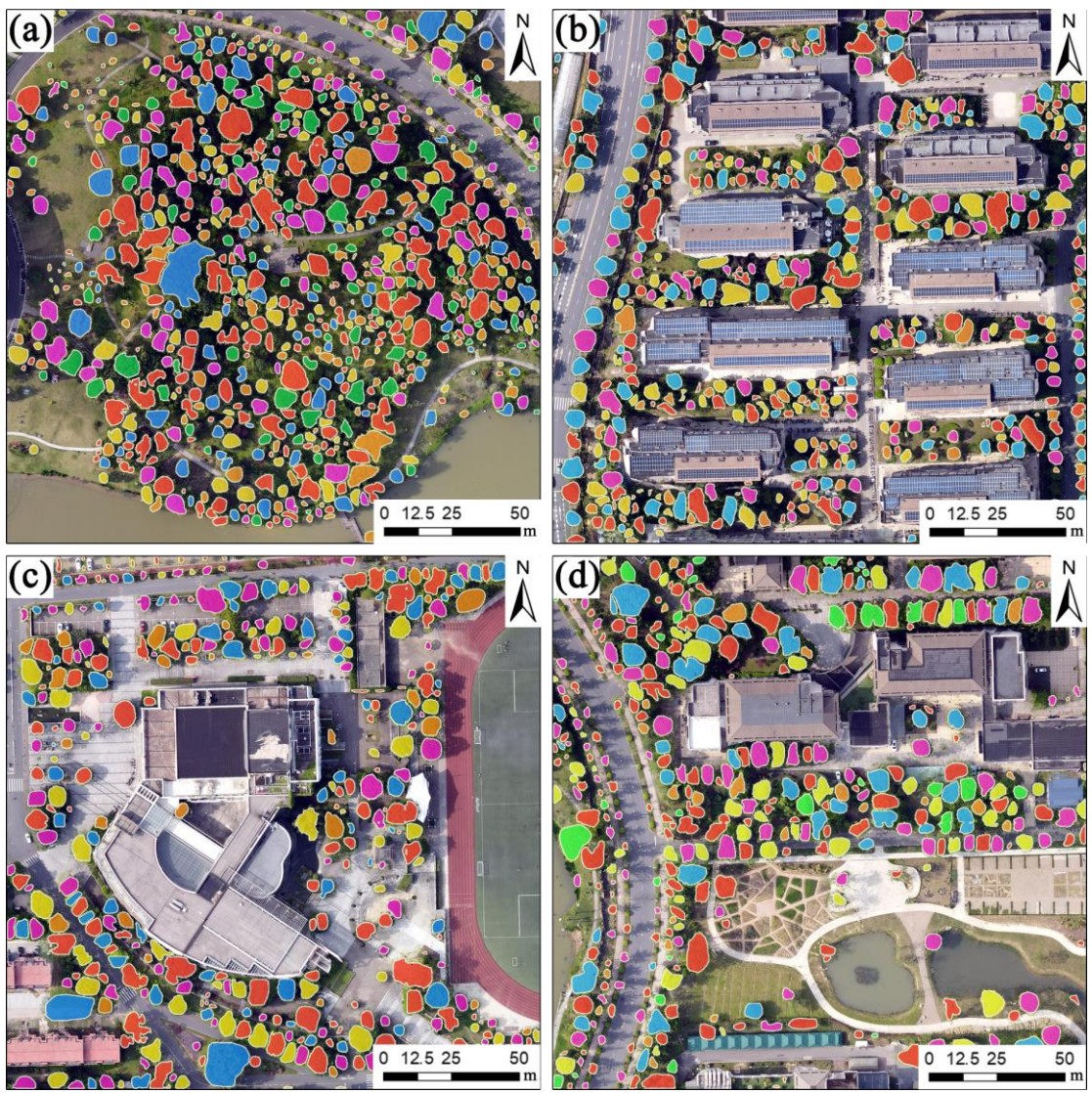

**Figure 10.** Urban forest multi-scene prediction results. The different colors were used to distinguish different individual trees. (**a**) Dense forest distribution, (**b**) Building forest intersection, (**c**) Active plaza vegetation, (**d**) Street trees.

**Table 5.** Statistics of the prediction results of improved model scenes.

| Scenario | Number of Tested Plants | Number of Real Plants | Number of Missed Plants | Number of Wrong Plants | Accuracy | Recall |
|---|---|---|---|---|---|---|
| Dense forest distribution a | 696 | 704 | 39 | 31 | 95.55% | 94.46% |
| Building forest intersection b | 421 | 409 | 22 | 34 | 91.92% | 94.62% |
| Active plaza vegetation c | 355 | 364 | 34 | 25 | 92.96% | 90.66% |
| Street trees d | 378 | 386 | 37 | 29 | 92.33% | 90.41% |
| Total | 1850 | 1863 | 132 | 119 | 93.57% | 92.91% |

## 4. Discussion

MCAN is based on the Mask R-CNN network framework with three improvements: first, the integration of CSPNet in the backbone feature extraction network to improve the feature extraction capability and reduce the missed detection of small targets; second, the use of the SiLU activation function to improve the nonlinearity and fitting capability of the

model; and third, the addition of a hybrid channel space attention mechanism in the FPN to improve the model's ability to recognize targets and remove background interference.

The original ResNet101 [65] backbone feature extraction network of the Mask RCNN network has a lengthy feature extraction path that loses a significant amount of spatial information and ignores the finer details of the image, which easily causes issues such as the missed or false detection of small targets [66] while also increasing the computational complexity and training difficulty. Compared with Resnet101, CSPResNet not only enhances the learning ability of the CNN but also reduces the size of the model so that it can maintain sufficient accuracy while being lightweight. This ensures the effectiveness and efficiency of the individual wood segmentation network. In addition, CSPResNet is able to improve the model's ability to capture image details and reduce the information loss in ResNet due to the deep network framework, making CSPResNet more suitable for remote sensing images with rich details.

Additionally, the SiLU activation function used in this network performs computations more quickly than other functions, such as ELU and GELU, and does not experience data overflow problems [54]. Compared to the ReLU activation function, the SiLU function's derivatives are smoother and continuous throughout the definition domain [67]. Additionally, by utilizing the SiLU function, the gradient on the deep model can be kept from vanishing, making it easier to extract features. In conclusion, adopting the SiLU activation function can enhance the model's nonlinearity and capacity to fit data, making it simpler to learn the precise details of the target shape and resulting in superior detection outcomes. Therefore, CSPResNet and SiLU activation functions have advantages in improving model representation, feature fusion, parameter reduction, maintaining gradient stability, and feature enhancement [54].

The attention mechanism emulates the cognitive awareness of humans, enabling a computer to zoom in on crucial details and prioritize the essential aspects of the data [68]. With the ongoing advancement and use of deep learning technology, attention mechanisms have been extensively used and investigated in a variety of contexts. For example, attention mechanisms are used to enhance the performances of neural networks in computer vision tasks, such as image classification, target detection, and image segmentation. The FPN of the Mask R-CNN network is fused without considering the feature map importance, which leads to the low accuracy of individual wood extraction in a complex background environment. Most of the existing methods assign the same weight to different channels when convolutionally extracting features without channel importance ranking. CA effectively compresses the feature dimensions and boosts the model's effectiveness by focusing greater attention on the channels containing key information while ignoring other noncritical channels [69]. Unlike conventional CNNs, which give different weights to each pixel in the feature map, SA focuses more on the key regions in the spatial location of the image and ignores some noncritical regions in the spatial location, which allows more attention to be focused on the pixels belonging to the foreground region of interest, thus reducing background interference and improving detection accuracy [38]. In this study, a SAM and CAM are fused in the FPN, which not only improves the differentiation of foreground and background in a complex urban forest environment but also increases the detection accuracy of small canopy individual wood and prevents the loss of information in the feature fusion process.

Although the accuracy of the method used in this work to identify individual trees in urban forests is high, there are still some drawbacks. The following areas will be addressed in future research: (1) To further validate and optimize the model's robustness and generalizability, we should collect more data containing complex scenarios and samples with complexity and label them accurately so as to reduce the false-positive rate and false-negative rate. (2) To further improve the model's accuracy, additional attention mechanisms and network framework improvement techniques should be considered. (3) This study area is an urban forest, and the training and testing of the network model is also based on urban forest UAV images. In addition, the stand structure and tree species of urban

forests are very different compared to natural forests, so the applicability and accuracy of the network for natural forests or other vegetation should be further discussed.

## 5. Conclusions

In this study, inspired by the Mask R-CNN network framework, we innovatively proposed the MCAN network and applied it to urban forest individual wood canopy detection. The results showed that (1) the individual tree detection AP value of the MCAN network is 92.40%, which is 3.7%, 3.84%, and 12.53% higher than that of Mask R-CNN, Faster R-CNN, and YOLOv5, respectively, demonstrating that the improved network can enhance canopy detection accuracy. (2) MCAN's segmentation AP value is 97.70%, which is 8.9% higher than that of Mask R-CNN, suggesting that the upgraded network's accuracy in canopy contour segmentation is higher than the original network, and the contour drawing error is lower. (3) By detecting the large-scale individual wood canopy in four typical scenes of urban forests, the accuracy rate is 93.57% and the recall rate is 92.91%, which indicate a good detection effect, demonstrating that the improved network has good detection capability for large scenes and is suitable for the segmentation and detection of individual wood in urban forests with high-resolution UAV remote sensing images.

**Author Contributions:** L.L.: Writing—original draft, Data curation, Methodology, Software, Validation, and Visualization. X.L., J.X. and F.M.: Writing—review and editing and Formal analysis. H.D.: Writing—review and editing, Conceptualization, Funding acquisition, Supervision, and Project administration. L.Z.: Writing—review and editing and Data curation. Y.Z.: Data curation. J.Y.: Data curation. L.H. and M.S.: Data curation. All authors have read and agreed to the published version of the manuscript.

**Funding:** The research was supported by the Leading Goose Project of Science Technology Department of Zhejiang Province (No. 2023C02035), the National Natural Science Foundation of China (No. 32201553 and 32171785), the Scientific Research Project of Baishanzu National Park (No. 2022JBGS02), the Talent launching project of scientific research and development fund of Zhejiang A & F University (No. 2021LFR029), and the Key Research and Development Program of Zhejiang Province (No. 2021C02005).

**Conflicts of Interest:** The authors declare that they have no known competing financial interest or personal relationships that could have appeared to influence the work reported in this paper.

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
