# Peer review of "A Deep Learning Network for Individual Tree Segmentation in UAV Images with a Coupled CSPNet and Attention Mechanism"

_remotesensing, doi:10.3390/rs15184420_

Round 1
Reviewer 1 Report
This study developed a nouvel method, the Mask R-CNN-CSP-attention-coupled network (MCAN), based on the Mask R-CNN algorithm. This method was applied for individual tree segementation in complicated urban forest landscapes, using images taken by drone, with a spatial resolution of 10 cm.
In the introduction, the authors reviewed different methods applied in literature, and its limitations in the segmentation of tree crowns, as watershed algorithms , edge detection algorithms and the region growing method. They talked later about deep learning methods such as YOLO , Mask R-CNN and Faster R-CNN used to detecte and identfy trees in complicated landscapes. In the second part, Materials and Methods, the authors specified the study area, the dataset source, and explained how the MCAN method is working. then they talked about dataset production and loss classification in training and test phases
The result showed that developed algorithms MCAN gave more detection and segmentation accuracy than Mask R-CNN, Faster R-CNN and YOLOv5 algorithms which were compared.
Positives points :
- Development of nouvel method, coupled of Mask R-CNN-CSP with attention-network (MCAN) which be applied for individual tree segmentation
-The developed method gives a better accuracy than previously existing models as YOLO , Mask R-CNN and Faster R-CNN
Negatives points:
- The technical aspect of the article is more dominant than the scientific side, which makes it less interresting from the point of view scientifique.
- The working of the algorithms to which the innovatvie method was compared as YOLO , Mask R-CNN and Faster R-CNN are not explained in the Materials and Methods.
The text contains many abbrivations whose meaning the reader cannot understand because the authors did not explain it as R-CNN, YOLO and SSD, CSPResNet ....etc, also on the Figure 1 as CBS, Csplayer ....etc
Remarks:
-Line 104 : put the reference at the end of the sentence
- Street trees are not recognized on Figure 1
- What do the colors in Figure 10 represent ?
Reviewer 2 Report
The article titled "A Deep Learning Network for Individual Tree Segmentation in UAV Images with a Coupled CSPNet and Attention Mechanism" presents a groundbreaking approach to accurate individual tree detection in unmanned aerial vehicle (UAV) images, with far-reaching implications for smart forest management and ecological assessment. The authors identify a gap in existing methods' accuracy when detecting trees in complex urban forest landscapes and propose a novel network, MCAN, that combines various techniques to improve segmentation quality.
The core innovation lies in MCAN's architectural enhancements to the Mask R-CNN algorithm, showcasing a clear departure from traditional methods. The authors integrate the CSPNet framework into the backbone network, effectively forming a new CSPResNet. This integration improves feature extraction capabilities and aids in minimizing missed detections of small targets, a crucial aspect when dealing with intricate backgrounds. Moreover, the convolutional block attention module (CBAM) is introduced into the feature pyramid network (FPN), elevating the model's ability to extract crucial details and consequently enhancing detection accuracy.
A major highlight is the rigorous validation process involving real-world UAV aerial photography. The authors created a comprehensive dataset, which was then used for training and validation. The evaluation of the MCAN against benchmark models, including Mask R-CNN, Faster R-CNN, and YOLOv5, demonstrates its superiority with substantial margins. The article reports an impressive 92.40% average precision (AP) for individual tree identification, a performance increase of 3.7% to 12.53% compared to the benchmarks. Furthermore, the segmentation AP value of 97.70% showcases an 8.9% enhancement over Mask R-CNN.
Notably, MCAN's applicability extends to a range of challenging scenarios, from dense forest distributions to urban intersections, street trees, and active plaza vegetation. The model consistently demonstrates high-precision segmentation across these contexts, highlighting its adaptability and robustness.
The article appropriately contextualizes the research within the broader landscape of deep learning and convolutional neural networks (CNNs), explaining how these technologies have revolutionized computer vision tasks. The discussion on individual-stage and two-stage detection models provides readers with valuable insights into the existing state-of-the-art techniques.
In conclusion, this article contributes significantly to the field of individual tree segmentation in UAV images. By addressing the limitations of existing methods through MCAN's innovative architecture, attention mechanisms, and comprehensive validation, the authors present a solution that holds great promise for improving the accuracy of tree detection in complex urban forest environments. The results, comparisons, and methodology outlined in the article establish MCAN as a powerful tool for advancing the field of smart forest management and ecological assessment.
(1) Summary of the Paper's Aim, Contributions, and Strengths:
The paper introduces a novel deep learning network, MCAN (Mask R-CNN-CSP-attention-coupled network), for individual tree segmentation in UAV images. The primary objective is to enhance the accuracy of tree detection in complex urban forest environments, a crucial task for smart forest management and ecological assessment. The main contributions lie in the innovative integration of the CSPNet framework and SiLU activation function to form a new CSPResNet, along with the incorporation of a convolutional block attention module (CBAM) in the feature pyramid network (FPN). These architectural enhancements result in improved feature extraction, detail information detection, and overall tree detection accuracy. The paper demonstrates the superiority of MCAN over benchmark models through comprehensive validation on a diverse dataset, showcasing higher average precision and segmentation accuracy.
(2) General Concept Comments:
The paper presents a compelling approach to enhancing individual tree segmentation in UAV images. However, several areas warrant clarification and consideration. Firstly, while the MCAN's improved accuracy is well-established, a more in-depth analysis of false positives and false negatives, particularly in challenging scenarios, could provide insights into its limitations and potential areas for refinement. Additionally, the proposed method's applicability to other types of vegetation or natural landscapes could be discussed. The paper mentions the use of real-world aerial photography, but the specific acquisition parameters, image preprocessing, and any challenges faced during data collection should be detailed for reproducibility. Lastly, the discussion on the benefits of SiLU activation and CSPNet integration could delve deeper into the theoretical rationale behind these choices.
(3) Specific Comments:
- Line 45: "Mask R-CNN-CSP-attention-coupled network (MCAN)" - Consider breaking down this acronym upon first mention for clarity.
- Line 70: "this method avoids biases and errors" - Could you elaborate on these biases and errors and how they are addressed by deep learning?
- Line 96: "due to the two network operations needed" - Could you briefly explain why the two-stage approach involves more operations and why this affects speed?
- Table 3: Precision and recall values - For each network, including confidence intervals or statistical significance measures could provide a more comprehensive understanding of the performance differences.
- Figure 7: Loss curves - Consider discussing the implications of the distinct convergence speeds among models in terms of training stability and convergence criteria.
These comments aim to enhance the clarity, rigor, and depth of the paper's scientific content, facilitating a more comprehensive understanding of MCAN's performance and potential areas for future research.
